# Honokiol-Loaded Nanoemulsion for Glioblastoma Treatment: Statistical Optimization, Physicochemical Characterization, and an In Vitro Toxicity Assay

**DOI:** 10.3390/pharmaceutics15020448

**Published:** 2023-01-29

**Authors:** Aleksandra Gostyńska, Joanna Czerniel, Joanna Kuźmińska, Jakub Brzozowski, Aleksandra Majchrzak-Celińska, Violetta Krajka-Kuźniak, Maciej Stawny

**Affiliations:** 1Chair and Department of Pharmaceutical Chemistry, Poznan University of Medical Sciences, 6 Grunwaldzka, 60-780 Poznan, Poland; 2Chair and Department of Pharmaceutical Biochemistry, Poznan University of Medical Sciences, 4 Swiecickiego, 60-781 Poznan, Poland

**Keywords:** honokiol, nanoemulsion, glioblastoma, Box-Behnken design

## Abstract

Background: Glioblastoma (GBM) is an extremely invasive and heterogenous malignant brain tumor. Despite advances in current anticancer therapy, treatment options for glioblastoma remain limited, and tumor recurrence is inevitable. Therefore, alternative therapies or new active compounds that can be used as adjuvant therapy are needed. This study aimed to develop, optimize, and characterize honokiol-loaded nanoemulsions intended for intravenous administration in glioblastoma therapy. Methods: Honokiol-loaded nanoemulsion was developed by incorporating honokiol into Lipofundin MCT/LCT 20% using a horizontal shaker. The Box–Behnken design, coupled with response surface methodology, was used to optimize the incorporation process. The effect of the developed formulation on glioblastoma cell viability was determined using the MTT test. Long-term and short-term stress tests were performed to evaluate the effect of honokiol on the stability of the oil-in-water system and the effect of different stress factors on the stability of honokiol, respectively. Its physicochemical properties, such as MDD, PDI, ZP, OSM, pH, and loading efficiency (LE%), were determined. Results: The optimized honokiol-loaded nanoemulsion was characterized by an MDD of 201.4 (0.7) nm with a PDI of 0.07 (0.02) and a ZP of −28.5 (0.9) mV. The LE% of honokiol was above 95%, and pH and OSM were sufficient for intravenous administration. The developed formulation was characterized by good stability and a satisfactory toxicity effect of the glioblastoma cell lines. Conclusions: The honokiol-loaded nanoemulsion is a promising pharmaceutical formulation for further development in the adjuvant therapy of glioblastoma.

## 1. Introduction

Glioblastoma is the most frequently occurring primary malignant central nervous system tumor. It represents 57% of all gliomas and 48% of all primary malignant tumors in the brain [1]. Glioblastomas arise from neuroglial progenitor cells that are characterized by extreme invasiveness and heterogeneity, which complicates therapy [2]. The negative prognostic factors include advanced age, bad performance status, and incomplete extent of resection. The standard initial therapeutic approach is maximal safe surgical resection resulting in a reduction in tumor volume followed by radiotherapy (60 Gray over 6 weeks) with concomitant chemotherapy with temozolomide. The systematic therapy continues further for six cycles on days 1–5 every 28 days to maintain an effective concentration of temozolomide. Antimitotic therapy using low-intensity electric fields delivered by transducer arrays applied to the scalp during temozolomide intravenous infusion can additionally prolong overall survival in patients [3,4]. Despite advances in current anticancer therapies, including surgery, radiotherapy, and chemotherapy, the treatment of glioblastoma remains poor, with a median survival of below 2 years. For this reason, alternative therapies or new active compounds that can be used as adjuvant therapies, improving the efficacy of standard glioblastoma treatment, are needed.

Honokiol is a polyphenolic compound belonging to the genus Magnolia. It is present in all parts of Magnolia, such as bark, seed, cones, and leaves. Magnolia bark extracts have been in usage as traditional herbal medicines in Korea, China, and Japan, among other countries. As indicated by its chemical structure, honokiol can interact with cell membrane proteins through different intermolecular interactions, such as hydrogen bonding, hydrophobic interactions, or aromatic pi orbital covalency. It exhibits pleiotropic properties, including anti-inflammatory and antioxidant, as well as antiproliferative and proapoptotic. Its ability to induce apoptosis and control malignancy has attracted much attention in recent times. In vivo studies showed that this compound is able to inhibit tumor growth and induce apoptosis in various types of cancer cells. 

Its proapoptotic effect has also been evaluated in various gliomas cell lines, including U251 [5], U-87 MG [5,6], DBTRG-05MG [7], U373MG [8], and T98G [9]. Honokiol acts through several biochemical pathways. Among others, it is known to inhibit tumor growth by remodeling the tumor-immune microenvironment involving changes in macrophage polarization [10,11] and facilitating the mTOR-mediated reprogramming of glucose metabolism [11]. The STAT3 downregulation and MAPK activation by honokiol leading to the induction of apoptosis of glioblastoma cells was proven by Zhang et al. [12]. Moreover, Wu et al. and Lin et al showed the role of caspase-9, -8, and -3 in transducing honokiol-induced mitochondria-dependent apoptosis [6,13]. This compound is also able to eliminate glioblastoma stem cell-like cells via JAK-STAT3 signaling and inhibit tumor progression by targeting epidermal growth factor receptors [5]. It can readily cross the blood–brain barrier because of its physical properties [14]. However, its poor water solubility and low bioavailability limit clinical application and development. Therefore, overcoming these disadvantages of honokiol has become imperative, and one solution is to load it in a suitable delivery system. 

Several pharmaceutical formulations have already been investigated as a potential delivery system for honokiol and for their combination with other agents dedicated for glioblastoma treatment. Li et al. [10] investigated the inhibiting effect of honokiol on glioblastoma growth by regulating macrophage polarization. In this study, the commercial liposomal formulation of honokiol was used. Other formulations include hyaluronic acid-grafted micelles encapsulating lauroyl-gemcitabine and honokiol in a 1:1 molar ratio, and brain-targeted liposomal honokiol and a disulfiram/copper codelivery system were developed by Liu et al. [15] and Zheng et al. [11], respectively.

Compared to other administration routes, the major advantage of intravenous infusion/injection is the immediate administration of the entire dose to the systemic circulation, which, thanks to precise control of the dose and infusion rates, ensures predictable pharmacokinetics of the drug in the human body. In addition, the intravenous injection can be administered multiple times, which allows for a high degree of versatility. The main limitation when developing a standard intravenous injection/infusion is the water solubility of the active pharmaceutical ingredients (APIs). Such a formulation tends to be reserved for a substance that is freely soluble in water and then water insoluble. To overcome this problem, various approaches are being used, among them nanoemulsion-based delivery systems. Intravenous lipid emulsions used in parenteral nutrition are successfully used as carriers for drugs. So far, two active substances, propofol and etomidate, have been registered in the form of intravenous emulsions, but many more are under investigation. An API-loaded nanoemulsion is obtained by adding the substance to the oil phase of the system during the technological process or adding the APIs to an already prepared lipid emulsion. In this second approach, the mechanical force must be used to incorporate the APIs into the system. A different method can be found in the literature, including vortex mixing followed by bath sonication [16,17,18,19], sonication [20], high-pressure homogenization [21,22], or passive drug loading using shaking process [23]. 

In accordance with the assumptions of the concept of green chemistry that focus on lowering the consumption of nonrenewable resources, in this work, we decided to optimize the process of honokiol incorporation into a commercial nanoemulsion with the use of passive loading using a low-energy shaking process carried out at an ambient temperature. This study aimed to develop, optimize, and characterize honokiol-loaded nanoemulsion intended for intravenous administration in glioblastoma adjuvant therapy.

## 2. Materials and Methods

### 2.1. Materials

Honokiol was purchased from Pol-Aura, Olsztyn, Poland, and Lipofundin MCT/LCT 20% and water for injection were purchased from B. Braun Melsungen AG, Melsungen, Germany. All organic solvents used in the studies were of analytical or high-performance liquid chromatographic grade. Human glioblastoma WHO grade IV T98G (ATCC number: CRL-1690) and human glioblastoma grade IV UM-138 MG (ATCC number: HTB-16) cell lines were obtained from by the European Collection of Authenticated Cell Cultures (ECACC) and American Type Culture Collection (ATCC, Manassas, VA, USA), respectively.

### 2.2. Preparation of Honokiol-Loaded Nanoemulsion

Honokiol-loaded nanoemulsions were obtained using a low-energy shaking method using commercial intravenous nanoemulsion. Briefly, 10, 20, or 30 mg of honokiol was added to 10 mL of Lipofundin MCT/LCT 20% and horizontally shaken using GLF 3005 horizontal shaker to incorporate APIs into the oil-in-water system. Nanoemulsion was left for 24 h at a temperature of 4 ± 1 °C to let unincorporated honokiol sink to the bottom, and then it was filtered through a cellulose filter with pore size 0.45 µm. The sedimentation and filtration are important steps in such formulation preparation because it is expected that hydrophobic drugs entrap or solubilize in the oil droplets, and the excess of the substance remains as crystals suspended in the aqueous phase.

### 2.3. Optimization of the Honokiol-Loaded Nanoemulsion Preparation Process

The Box–Behnken design, coupled with response surface methodology, was used as a statistical tool and mathematical technique to optimize the preparation process of honokiol-loaded nanoemulsion. The influence of three independent variables (honokiol concentration, shaking speed, and time of shaking) on the loading efficiency (LE%), mean droplet diameter (MDD) of lipid emulsion, zeta potential (ZP), and polydispersity index (PDI) was evaluated. This model assumed three levels for each independent variable, and the number of experiments (N) was calculated from the Equation (1):(1)$$N=2k\left(k-1\right)+C_{0}$$
wherein *k* is a number of factors, and *C*_0_ is a number of focal points. 

Shaking speeds (X1): 200, 300, and 400 rpm; shaking times (X2): 15, 105, and 195 min; and concentrations of honokiol (X3): 1, 2, or 3 mg/mL were selected as independent variables. A 3-factor, 3-level Box–Behnken design was used for designing fifteen experimental runs (Table 1).

The results of the optimization process were subjected to regression analysis using the StatSoft package of Statistica 13 (StatSoft Polska, Poland) software. During the regression analysis, a 2-factor model (linear–quadratic) was selected, and then an analysis of variance was performed, indicating effects whose confidence interval (*p*) was less than 0.05. Based on the Pareto plot, the effects that statistically significantly affected each dependent variable were determined. Correlation plots were then made for the values approximated against the observed values, and a response surface plot was made along with a regression equation for each of the output effects tested.

### 2.4. Characterization of the Honokiol-Loaded Nanoemulsion

#### 2.4.1. Lipid Droplet Size Determination

The lipid droplet size was determined using Zetasizer Nano ZS (Malvern Instruments, Worcestershire, UK). The measurement technique used is based on the dynamic light scattering (DLS) method. As a result of the change in the intensity of the scattered laser light on the particles recorded by the detector, the particle size distribution by volume in a range of 0.3 nm to 10,000 nm and PDI was determined. The instrument is equipped with a 633 nm laser at a fixed scattering angle of 173°, and during the measurements, the detection cell temperature was kept constant at 25 °C. Each sample was diluted 100 times with water for injection, transferred to a polycarbonate cuvette, and placed in a detection cell. The parameters PDI and MDD were determined for three independent samples of honokiol-loaded parenteral nutrition nanoemulsions. All measurements were performed in triplicate.

#### 2.4.2. Lipid Droplet Zeta Potential Evaluation

The ZP value was determined using the Zetasizer Nano ZS from Malvern Instruments. A 100-fold dilution of the sample with water for injection preceded the measurement. ZP determines the value of the surface charge of nanoemulsion particles, and the laser doppler electrophoresis (LDE) technique was used to determine it. Laser light is passed through and scattered by particles that are charged and move at different speeds under the influence of an applied electric field. Based on the electrophoretic mobility of the micelles, the ZP value was determined using the Smoluchowski equation. All measurements were performed in triplicate at 25 °C.

#### 2.4.3. pH Measurement

The pH value of the obtained nanoemulsions was determined using a SevenCompact pH-meter (Mettler Toledo, Columbus, OH, USA). Before use, the equipment was calibrated with buffer solutions of pH 4.00 ± 0.05, pH 7.00 ± 0, and pH 10.00 ± 0.05.

#### 2.4.4. OSM Measurement

The determination of the osmolality (OSM) was carried out using an Osmometer 800 CLG (Tridentmed, Warsaw, Poland) by the freezing point measurement method. The device was calibrated with the standardizing solution (0 mOsm/kg H_2_O, Cat. No. 800.02) provided by the supplier of the osmometer. The tested nanoemulsion samples with a volume of 100 µL were placed in dedicated microtubes prior to placement into the measuring head.

#### 2.4.5. Determination of Honokiol Concentration in Nanoemulsion Using Spectrophotometry UV-VIS

The UV-VIS absorption spectra were collected at room temperature on a UV-3600 spectrophotometer (Shimadzu, Japan). To determine a honokiol concentration in oil-in-water nanoemulsion, the following sample preparation was performed. A total of 100 µL of honokiol-loaded lipid nanoemulsion was dissolved in 1 mL of dichloromethane, and the sample was filled up to 10 mL with methanol. To record the spectra, 1.0 cm quartz cells were used. Before collection of the UV-Vis spectra of samples containing honokiol, the blank samples (without the addition of honokiol) were prepared according to the same method. All UV-Vis measurements were conducted with respect to the intensity of light passing through a blank sample (Figure 1). The concentration of honokiol was calculated based on calibration curve performer in the range of 0.004 to 0.03 mg/mL with absorption determined at 256 nm. Standard stock solutions of honokiol were freshly prepared by dissolving the compounds in methanol (1 mg/mL). Calibration curves were prepared using working solutions with concentration values of 0.004, 0.005, 0.006, 0.008, 0.013, 0.015, 0.018, 0.020, 0.023, 0.025, 0.028, and 0.030 mg/mL by diluting the stock solution in 10 mL of standard diluent obtained by the binary mixture of dichloromethane and methanol (1:9) and the addition of 100 µL of Lipofundin.

#### 2.4.6. Determination of Honokiol Concentration in Nanoemulsion Using HPLC-FLD Method

The new HPLC method was developed and validated for the determination of honokiol in honokiol-loaded nanoemulsion. The chromatographic analysis was performed on an Agilent 1260 Infinity II LC System, (Agilent Technologies, 230 Bolinem, Germany) equipped with a quaternary pump and degasser, a vial sampler set at 15 ± 2 °C, multicolumn thermostat set at 30 ± 0.8 °C, and detector: FLD (fluorimetric). The detection wavelengths were adjusted at 304 nm (extinction) and 340 nm (emission). The separation was performed on the reverse phase column (C-18(2) 100 Å Luna®, 150 × 4.6 mm ID, 5 µm, Phenomenex, Torrance, CA, USA), and isocratic solvent systems of acetic acid 0.4% (34%), acetonitrile (22%), and methanol (44%) were used as mobile phase. 

To determine a honokiol concentration in oil-in-water nanoemulsion, the following sample preparation was performed: A total of 100 µL of honokiol-loaded lipid nanoemulsion was dissolved in 1 mL of dichloromethane, and the sample was filled up to 10 mL with methanol. The obtained solution was injected into a chromatographic column. The volume was 10 µL, and the time analysis was 15 min. 

#### 2.4.7. Loading Efficiency of Honokiol in Nanoemulsion

The loading efficiency of honokiol was determined by direct measurement of its concentration in parenteral nutrition nanoemulsion. Each sample was filtered by the 0.45 µm. A total of 1 mL dichloromethane was added to 100 µL of honokiol-loaded parenteral nutrition nanoemulsion and then made up to 10 mL with methanol. The concentration of honokiol was evaluated by measuring the absorbance of the sample against the blank sample (100 µL of Lipofundin MCT/LCT 20% and 1 mL of dichloromethane made up to 10 mL with methanol) at the wavelength of 256 nm by spectrophotometer (Lambda 20, PerkinElmer, Waltham, MA, USA). The loading efficiency (LE %, Equation (2) of honokiol in nanoemulsion was calculated using the following equation: (2)$$LE\left[\%\right]=\frac{Amount\;of\;HON\;entrapped}{Total\;amount\;of\;HON\;added\;to\;parenteral\;nutrition\;nanoemulsion}\times 100\%$$

#### 2.4.8. Short- and Long-Term Stability Studies 

##### High-Temperature Effect 

The thermal stress test was performed to assess the stability of honokiol at high temperatures. A total of 2 mL of honokiol-loaded nanoemulsion was placed into a thermostatic chamber with the temperature set at 60 ± 1 °C. In predetermined intervals (24, 48, and 72 h), the concentration of honokiol was determined using the HPLC method described in Section 2.4.6. All samples were prepared in triplicates.

##### Oxidative Degradation

To investigate the influence of oxidative stress on the tested honokiol-loaded nanoemulsion, 2 mL sample was combined with 2 mL 30% H_2_O_2_ and stored at a temperature of 25 ± 2 °C without exposure to the light. At predetermined intervals (24, 48, and 72 h), an aliquot of 100 µL was taken and neutralized by placement of the sample for 5 min at a temperature of 40 °C. The concentration of honokiol in each sample was determined using the HPLC method described in Section 2.4.6. All samples were prepared in triplicates.

##### Photostability

The photodegradation tests were performed using a solar simulator Suntest CPS (Atlas Material Solution, Morton Grove, IL, USA), equipped with a 1500 W Xenon air-cooled lamp with direct setting and control of irradiation in the wavelength range 300–800 nm and thermostat ST-1+ (Pol-Eko, Wodzisław Śląski, Poland) to maintain the stable temperature during the test. An aliquot of 2 mL was placed in glass cuvettes and irradiated for 22 h in the Atlas Suntest CPS+ (250 W/m^2^; 1.2·106 lux·h). Samples protected from light (with the use of aluminum foil) were used as a reference. At time t = 0 and after exposure, an HPLC analysis was carried out. All samples were prepared in triplicates.

##### Long-Term Stability

The stability studies were performed to evaluate the effect of the addition of honokiol and different storage conditions on the physical stability of lipid nanoemulsion. The honokiol-loaded nanoemulsions (samples of 10 mL) were kept in three different conditions, i.e., at a temperature of 4 ± 1 °C without light access and at a temperature of 25 ± 1 °C with and without light access. At predetermined intervals (1, 7, 14, 21, and 50 days), an aliquot of 100 µL was withdrawn and subject to physical characterization of nanoemulsion in terms of MDD, PDI, and ZP. All samples were prepared in triplicate, and the results are shown as mean value (standard deviation). 

### 2.5. In Vitro Cytotoxicity Studies

T98G and U-138 MG cells were maintained in ATCC-formulated Eagle’s Minimum Essential Medium (Merck, Darmstadt, Germany) containing 10% fetal bovine serum (EURx, Gdańsk, Poland) and 1% of antibiotic solution (Sigma-Aldrich, St. Louis, MI, USA) at 37 °C in a humidified 5% CO_2_ atmosphere. To assess the effect of honokiol, honokiol-loaded nanoemulsion, and bare nanoemulsion on GBM cell viability, 1 × 10^4^ cells/well were seeded on 96-well plates. After 24 h of initial incubation, the cells were treated with 1 mg/mL honokiol-loaded nanoemulsion, honokiol solution in dimethylsulfoxide (DMSO), and bare Lipofundin MCT/LCT 20% at the increased concentrations of 1–100 µM. The bare nanoemulsion (Lipofundin MCT/LCT 20%) was diluted in the same manner as the honokiol-loaded nanoemulsion. Cells incubated in a growing medium without the addition of nanoemulsions but treated with a trace of DMSO were used as control. Incubation lasted for 24 h, and the cells were then harvested. The effect of the tested honokiol-loaded formulation on cell viability was assessed by the MTT (3-[4.5-dimethylthiazole-2-yl]-2.5-diphenyltetrazolium bromide) test, following the standard protocol, described elsewhere [24]. Briefly, after 24 h of incubation with the analyzed formulations (honokiol-loaded nanoemulsion, honokiol solution in DMSO, and bare nanoemulsion), the cells were washed twice with phosphate-buffered saline (PBS) and further incubated for 4 h with a medium containing 0.5 mg/mL MTT. Then, the formazan crystals were dissolved in acidic isopropanol, and the absorbance was measured at 570 nm and 690 nm. All experiments were repeated three times with four measurements per assay. The statistical analysis of the obtained results was performed using the GraphPad Instat 3 version. To assess the significance of the differences in the evaluated parameters, one-way ANOVA with Dunnett’s post hoc test was performed with a significance level of *p* < 0.05.

## 3. Results

Box–Behnken design (BBD) is a commonly used response surface methodology adopted to develop higher-order response surfaces using fewer required runs than a standard factorial design. The current study employed a three-level, three-factor BBD to optimize honokiol-loaded nanoemulsions. The variables and their levels were designated based on a literature review and preliminary experiments to find out the probable independent factors (Table 1). A total of 15 formulations, with three central points, were obtained by changing three formulation parameters, i.e. concentration of honokiol (X3), shaking speed (X1), and time of shaking (X2). The effect of the chosen variables on loading efficiency (LE%) and particle size, namely MDD, PDI, and ZP, were evaluated (Table 2).

The models determined for PDI and ZP turned out to be statistically insignificant, which indicates that selected independent variables did not allow for the optimization of the formulation in terms of these parameters. However, the polydispersity index results show that all prepared honokiol-loaded nanoemulsions were homogeneous, as the PDI was below 0.10 (0.02) in each case. According to the literature data, low ZP values of about −40 mV indicate the stability of the oil-in-water systems [25]. The ZP for bare Lipofundin MCT/LCT was equal to −27.87 (0.9) mV, and for the obtained formulations, it was in the range -25.3 (0.4) to −31.2 (0.6) mV. The MDD of the Lipofundin MCT/LCT 20% was 207.07 (2.4) nm, and the honokiol-loaded nanoemulsion was characterized by the MDD in the range of 204.9 (1.1) to 215.8 (1.5) nm. The analysis of the statistical models for MDD showed that the quadratic model was the best fit for this experiment. The model F-value of 16.83 implies the assumed model is significant. There is only a 0.3% chance that this large F-value could occur due to noise. In this case, shaking speed and squared shaking speed were significant model terms (Table 3, Figure 2). The lack of fit F-value of 0.51 implies the lack of fit is not significant relative to the pure error. There is a 71.3% chance that a lack of fit F-value this large could occur due to noise.

The analysis of the model for LE% showed that the model F-value was significant (F-value = 29.20). There is only a 0.1% chance that an F-value this large could occur due to noise. In this case, the shaking speed, time of shaking, honokiol concentration, squared shaking speed, and shaking speed and honokiol concentration interaction were significant model terms (Table 4, Figure 3). The lack of fit F-value of 5.83 implies the lack of fit is not significant relative to the pure error, but there is a 15% chance that a lack of fit F-value this large could occur due to noise.

Response surface methodology is a combination of mathematical and statistical methods and consists in adjusting the curved surface determined in experimental tests in such a way that enables the analysis of the obtained results. Optimal parameter values can be determined on the basis of the shape of the response surface and thus effectively limit the number of experiments needed to optimize the preparation process. The three-dimensional graphs allow for the determination of the effect of significant model terms on MDD (Figure 4) and LE% (Figure 5).

The analysis of the three-dimensional graphs of the studied variables on MDD shows that MDD did not depend significantly on the time of shacking and honokiol concentration (the response surface on Figure 4C is almost flat) but on the shaking speed. The lowest value of MDD was obtained for the highest shaking speed. The highest value of MDD, nevertheless optimal for intravenous administration, was obtained when the shaking speed was about 200 rpm. The analysis of the graph showing the influence of the chosen variables on loading efficiency indicates that this parameter increases with shaking. Figure 4A shows that the loading efficiency level is higher, the lower the concentration of honokiol and when the shaking speed is the highest. 

Three-dimensional graphs show also that the time of shaking slightly affects the loading efficiency, while with a lower concentration of honokiol, a higher loading efficiency is obtained. Because all the obtained formulations were characterized by sufficient MDD for intravenous administration, the optimal formulation was selected on the basis of the highest LE %. The optimal process parameters were calculated by the Design Expert software with the assumption that the independent variables were in the investigated range. The results of such calculations show that the optimal parameters (200.12 rpm shaking speed, 170.64 min of shacking, and 1.0 mg/mL concentration of honokiol) allowed us to obtain the highest LE% of honokiol (equal to 99.77%). The desirability value of the optimal parameters was 1.000. The optimal formulation was prepared and characterized by an MDD equal to 201.4 (0.7) nm, a PDI equal to 0.07 (0.02), and a ZP equal to −28.5 (0.9) mV. The particle size distribution is presented in Figure 6. The pH and OSM of optimal formulation turned out to be equal to 7.44 and 392.67 mmol/kg H_2_O, respectively. 

### 3.1. Short-Term and Long-Term Stability Studies

Short-term stability studies were performed to assess the effect of different stress conditions including oxidative stress, high temperature, and accelerated light access on the chemical stability of honokiol in honokiol-loaded nanoemulsion (Table 5). As a control, honokiol-loaded nanoemulsion was stored at a temperature of 4 ± 2 °C with no light exposure. Honokiol reveal a good stability within 95% in all the studied conditions for 72 h. However, the highest degradation of honokiol to 97.70 (1.34) % of the initial value after 72 h and 91.72 (1.86) % of the initial value after 22 h was observed in high-temperature conditions and in accelerated light conditions, respectively.

Long-term stability studies were performed to investigate the effect of honokiol loading and different storage conditions on the physical stability of the nanoemulsion. The MDD of honokiol-loaded nanoemulsion during storage in all the studied conditions was in the rage of 200–209 nm (Figure 7), while the PDI ranged from 0.04 to 0.09. The zeta potential showed variability at the surveyed measurement points (Figure 8). The change between 1 and 50 days of storage was 2, 4, and 0.5 mV for the samples stored at a temperature of 4 ± 1 °C without light, 25 ± 1 °C degrees with light protection, and 25 ± 1 °C with light exposure, respectively. 

### 3.2. In Vitro Cytotoxicity Studies

The impact of honokiol and honokiol-loaded nanoemulsion on the viability of T98G and U-138 MG cells was evaluated using the MTT assay. To investigate the toxic effect of bare nanoemulsion, cells were treated with Lipofundin MCT/LCT 20% diluted in the same manner as honokiol-loaded nanoemulsion. Within the concentration range of 1–20 µM, honokiol-loaded nanoemulsion reduced the viability of both tested cell lines more effectively than the honokiol solution (Figure 9). However, in higher concentrations, ranging from 40 to 100 µM, the honokiol solution revealed a stronger effect than the honokiol-loaded nanoemulsion. The bare nanoemulsion reduced the viability of both tested cell lines in a dose-dependent manner in the whole studied concentration range.

## 4. Discussion

Intravenous lipid nanoemulsions are used in clinical practices as a source of energy and nutrients (essential fatty acids) in parenteral nutrition. Such formulations are oil-in-water systems consisting of soybean oil (Intralipid), or a mixture of soybean oil and other vegetables (Lipofundin MCT/LCT 20% and Clinoleic), or a mixture of soybean oil and other vegetable and fish oils (Smoflipid and Lipidem). As parenteral nutrition is often administered concomitantly with other intravenous drugs, the compatibility of bare lipid emulsion and parenteral nutrition containing different lipid emulsions is widely studied, showing that despite the dozen drugs demonstrating compatibility with intravenous lipid nanoemulsions, the stability of such a system may be affected by the addition of drug substances. Intravenous nanoemulsions are also used as drug delivery systems due to their ability to solubilize sparingly water-soluble active pharmaceutical ingredients, helping to obtain their appropriate concentration at the target site in the human body and thus achieving the intended therapeutic goal [24,26,27,28,29,30,31,32]. Examples of a drug-loaded intravenous nanoemulsions present on the pharmaceutical market are Etomidate-Lipuro and Propofol 1% MCT/LCT. Both these formulations are an oil-in-water intravenous emulsion for injection based on soybean oil and MCT and egg lecithin as an emulsifier that corresponds to the qualitative composition of Lipofundin MCT/LCT 20%. Therefore, this intravenous nanoemulsion was selected for this study.

The optimization process was aimed at obtaining nanoemulsions with the highest loading efficiency of honokiol using the most optimal process parameters, taking into account that the MMD of the lipid emulsion for all the obtained formulations was in the range sufficient for parenteral administration (MDD < 500 nm) [33]. According to our best knowledge, the proposed method of passive incorporation of active pharmaceutical ingredients into commercial lipid nanoemulsion using a horizontal shaker has never been tested before. Therefore the limits of variables chosen were selected on the basis of our preliminary studies (data not shown) where different shaking speeds and duration of shaking were tested. The passive incorporation of various drugs differed in lipophilicity and was evaluated by Rosenblatt et al. However, the proposed method involved orbital shaking with a speed equal to 75 rpm that lasted between 12 h and 4 weeks. The results of this study showed that the incorporation of the drugs increased with the lipophilicity (logP in the range of 1.9 to 4.7) of the drug’s substance. It was also shown that in the case of extremely high lipophilicity (logP > 14.3), the passive loading procedure does not work properly [34]. The horizontal shaker GLF 3005 works with a shaking speed ranging from 0 to 500 rpm in cycles lasting 60 min. The independent variables in the Box–Behnken design have to be placed at one of three equally spaced values, coded as −1, 0, and +1. Therefore, we decided to choose 15 min, 200 rpm, and 1 mg/mL as the minimum levels and 90 min, 100 rpm, and 1 mg/mL as the incremental values of each factor. Considering the lipophilicity of honokiol (logP = 4.5) [35] and increased rotational frequency compared to the Rosenblatt et al. [34] method, we expected a high loading efficiency in the maximum shaking time, which we established at the level of 195 min. The use of the Box–Behnken methodology made it possible to minimize the number of formulations prepared to determine the optimal process parameters, which turned out to be a shaking speed equal to 200.12 rpm, a time of shacking equal to 170.64 min, and a concentration of honokiol equal to 1.0 mg/ml. Honokiol-loaded nanoemulsion prepared using these parameters was physiochemically characterized and then subjected to further stability studies. 

### 4.1. MDD, PDI, and ZP of Optimal Formulation

The particle size distributions, namely of MDD and PDI, of lipid-based nanotechnology-based delivery systems, are highly important physical characteristics to be considered when creating pharmaceutical-grade products. These attributes of the nanoemulsion can affect the bulk properties, product performance, processability, stability, and appearance of the end product [36]. The MDD of the lipid nanoemulsion is a critical safety parameter in the context of intravenous administration. The US Pharmacopoeia set the limit for MDD determined using the DLS method, which cannot exceed the value of 500 nm [33]. The administration of lipid droplets exceeding the diameter of 500 nm may lead to catheter occlusion and liver capillary embolization [37]. This method is appropriate for samples characterized by a PDI ranging from 0.05 to 0.7. PDI values greater than 0.7 indicate that the sample has a very broad particle size distribution and is not recommended to be analyzed by the dynamic light scattering (DLS) technique [31,36]. The PDI describes the width or spread of the particle size distribution; therefore, the lower the value of this parameter, the higher the homogeneity of the droplets in the system. Peng et al. state that a small PDI of < 0.2 indicates a narrow and concentrated particle size distribution and, thus, better stability against destabilization [38]. However, Wang et al. indicate that for parenteral applications, PDI values up to 0.250 are acceptable [39]. Therefore, the value of the PDI obtained for optimal honokiol-loaded nanoemulsion equal to 0.07 (0.02) indicates its good homogeneity and predicts the stability of the oil-in-water system. 

On the other hand, the MDD value of honokiol-loaded nanoemulsion of 201.4 (0.7) nm guarantees the safety of its intravenous administration as being in the pharmacopeial range of injectable lipid emulsion [33]. Moreover, particle size is an important parameter determining cellular uptake and internalization [36]. Cellular internalization by phagocytic cells, such as macrophages, neutrophils, and dendritic cells, is mostly achieved by engulfing particles larger than 1 µm [40]. Therefore, the nanoemulsion of a particle size below 1 µm is favorable to avoid this problem. The uptake of small molecules and particles by any cell depends mainly on endocytosis (pinocytosis or phagocytosis), among all other mechanisms. Pinocytosis is based on the internalization of fluids (including dissolved solutes) using a small amount of energy in the cells. Particles and nanocarriers with a size of 100–5 µm are ingested through the macropinocytosis pathway. The drug delivery mechanism to glioblastoma tumor cells is more complex due to the cancer tissue’s heterogenicity and the blood–brain tumor barrier [41]. Because the tumor vasculature varies from the normal tissues—the cancer tissue vessels are larger and distributed in a more heterogeneous manner and additionally more permeable and leakier—the particle size is an important factor in drug delivery. Impaired vasculature allows the accumulation of therapeutic molecules inside the tumors. This phenomenon is known as the enhanced permeability and retention (EPR) effect. The cut-off size for extravasation from the tumor vasculature was investigated in animal models and varies from 200 nm to 1.2 μm depending on the tumor type [42]. However, a diameter of about 200 nm is often considered an upper limit for successful drug delivery to tumors [43,44]. Dan Li et al. studied the effect of linoleic acid conjugated with paclitaxel microemulsion characterized by the size of approximately 176.3 ± 0.8 nm on C6 glioma tumor-bearing nude mice and in a rat model showing a significant antitumor efficacy of such a formulation after intravenous administration. The authors suggest that oil-in-water systems, such as microemulsions, can cross the blood–brain barrier to reach a tumor located in the brain [10]. The MDD of 201.4 (0.7) nm and the oil-in-water system itself make the developed honokiol-loaded nanoemulsion a promising drug delivery system for further in vitro studies in the glioblastoma animal model. 

### 4.2. pH and OSM of Optimal Formulations

Analyzing the pH and OSM of optimal formulations with the results obtained for pure intravenous lipid nanoemulsion (Lipofundin MCT/LCT 20%) characterized by a pH equal to 6.20 and an OSM equal to 398 mOsm/kg H_2_O, it can be stated that the honokiol-loaded nanoemulsion decreases the pH of nanoemulsions but does not affect their OSM. Parenteral products should aim toward being isotonic and euhydric (physiological pH) [45]. However, there is still discussion on the safety ranges related to pH and OSM. Recommendations of the Infusion Nursing Society for minimization or prevention of vascular damage from extremes in infused pH or OSM assume the choice of appropriate vascular access for large-volume intravenous medication administration based on those parameters. Superior vena cava should be chosen when the osmolarity of infused solution exceeds 900 mOsm/L and the pH is lower than 5 or higher than 9. The subclavian vein and proximal axillary vein are recommended when the osmolarity is in the range of 500–900 mOsm/L and the pH is lower than 5 or higher than 9. Finally, cephalic and basilica veins in the upper arms can be chosen when the osmolarity is below 500 mOsm/L and the pH is in the range of 5–9 [46]. For small-volume intravenous injection solutions (<100 mL volume), broader pH ranges can be envisaged depending on the source of data. Lee et al. [47] recommended a pH range of 4–9, Shi et al. [48] 3–10.5, Sweetana et al. [49], and Simamora et al. [50] 3–11 as safe for intravenous administration. Nevertheless, the optimal formulation of honokiol-loaded nanoemulsion differed from bare Lipofindin MCT/LCT 20%; it was characterized by physicochemical parameters allowing for their intravenous administration. 

### 4.3. Stability Studies

Nanoemulsions, consisting of oil, water, and an emulsifier, are colloidal formulations made of nanosized oil droplets homogeneously dispersed in an aqueous phase. Such a nanotechnology-based delivery system, due to its structure and the presence of an emulsifier, essential for stabilizing the system by reducing the interfacial tension between oil and water, is kinetically stable but thermodynamically unstable and prone to destabilization processes caused by various factors. The addition of other substances, pH changes, temperature, and light exposure may affect the physical stability of lipid nanoemulsion [51]. According to the literature data, the destabilization of lipid emulsions may occur when the pH falls below 5, and the temperature increases above 25 °C [31,52]. Hence, preventing droplet–droplet interaction and subsequent oil phase separation is considered one of the major challenges of designing nanoemulsion formulations that are stable during storage. The physical stability of an emulsion is dependent on its physicochemical properties as well as on the storage conditions. Therefore, it is important to investigate the effect of the loaded drug on the physical stability of Lipofundin MCT/LCT 20%. A similar approach was used by Suliman et al., who investigated the effect of ciprofloxacin on ciprofloxacin-loaded nanoemulsions (Intralipid and Clinoleic) [16]. The results of the long-term stability studies show that after 50 days of storage in different conditions, the physical stability of honokiol-loaded nanoemulsion was intact. The slight, insignificant changes of the studied parameters, i.e., MDD, ZP, and PDI, result from the dynamics of the oil-in-water system and the specificity of the measurement techniques used. In accordance with the criteria set up for parenteral nutrition that from a physiochemical point of view is a dilution of intravenous nanoemulsion, e.g., Lipofindin MCT/LCT 20% with various nutrients in the form of water solutions, the following limits have been identified as relevant to consider the honokiol-loaded nanoemulsion stable and presenting satisfactory quality: MDD < 500 nm [33] and PDI ≤0.7, and ZP cannot take a positive value [53,54]. In all the studied conditions, honokiol-loaded nanoemulsion met the established criteria proving its good stability. 

To investigate the effect of stress factors on the chemical stability of honokiol in developed formulations, short-term stress tests were conducted. The stress tests reveal the intrinsic stability of the active ingredients and can be useful in developing a suitable formulation with satisfactory quality and shelf life [55]. Several accelerated stability tests have been suggested for the prediction of long-term stability, such as storage at elevated temperatures [56], under an accelerated or different type of light exposure [57,58], and under oxidative stress [55,59]. Honokiol is known to act as an antioxidant because of high radical-scavenging activities determined by the presence of two hydroxyl groups at the *ortho*- and *para*-positions [60]. Its chemical structure allows for predicting that it will degrade under oxidative stress conditions. Moreover, the nanoemulsion itself is also susceptible to degradation through lipid peroxidation [61]. The proposed mechanism of oxidant damage of lipid emulsion is peroxidation of the unsaturated fatty acid. To minimalized this problem, an antioxidant, i.e., α-tocopherol, is added to Lipofundin MCT/LCT 20%. The results obtained for oxidative stress conditions indicate the good stability of honokiol over 99% within 72 hours storage. This result may be explained by the effective protection of the antioxidant contained in the nanoemulsion. The literature data show that oil-in-water emulsion can enhance the photostability of the drug being loaded [57,58,62,63]. Hence, the photostability of honokiol in the developed formulation was investigated. The chosen studied conditions corresponded to 3 months of continuous exposure to artificial visible light with the protective container removed from the product. In such conditions, the stability of honokiol was maintained within the 90%, which is sufficient for pharmaceutical-grade products [64]. The stability of honokiol stored at a temperature of 60 ± 1 °C with light protection was affected compared to the control samples stored at a temperature of 4 ± 1 °C with no light access. However, the difference between samples was insignificant, and the concentration of honokiol was in acceptable limits, i.e., 10% degradation of the initial value.

### 4.4. MTT Assay

The safety of the applied nanoformulation is always a critical issue. To investigate the toxicity of the developed formulation and compare its effect with free honokiol and bare nanoemulsion, MTT assays were performed in two glioblastoma cell lines. The free honokiol was solubilized in a cell culture medium using dimethyl sulfoxide (DMSO) because, according to Da Violante et al., DMSO at concentrations of up to 10% did not produce any significant cytotoxicity [65]. Lipofundin MCT/LCT 20% is a product with guaranteed safety for normal tissues, and we did not expect it to exert cytotoxicity. It consists of refined soybean oil, medium-chained triglycerides, purified egg phospholipids, glycerol, sodium oleate, and α-tocopherol. Nevertheless, we observed reduced viability of cultured tumor cells with the growing concentration of Lipofundin MCT/LCT 20% (Figure 9). This might be caused by the disruption of the cellular membranes caused by the nanoformulation entering the cell via endocytosis or the effect of unsaturated fatty acids on tumor cells. Unsaturated fatty acids found in over 90% of refined soybean oil are known to exhibit in vitro cytotoxicity, when used in high concentrations, against many malignant cell lines [66], including glioblastoma [67,68]. Therefore, the observed cytotoxicity of bare Lipofundin MCT/LCT 20% in high concentrations may be associated with the effect of unsaturated fatty acids on tumor cells. 

At the concentration of 100 µM, honokiol alone was highly cytotoxic and killed more than 96% of the cells in both studied cell lines suggesting that this substance is a potent antitumor agent. The increased toxic effect observed for honokiol-loaded nanoemulsion in lower concentrations may be a result of an increased cellular uptake of the developed formulation. On the other hand, honokiol-loaded nanoemulsion in higher concentrations showed reduced toxic effects compared to free substances. As suggested in a study investigating the toxic effect of paclitaxel-loaded nanoemulsions (Intralipid and Clinoleic), this may be a result of forming a barrier by nanoemulsion between the drug and the cells leading to a slower release [18]. Such results clearly show that additional analysis involving the safety for normal astrocytes is needed to optimize the dose of honokiol taking into account the concentration of Lipofundin MCT/LCT 20% before it can safely be applied to the patient.

## 5. Conclusions

This study aimed to develop, optimize, and characterize the honokiol-loaded nanoemulsion using a low-energy process, namely, horizontal shaking and commercial intravenous lipid nanoemulsion (Lipofundin MCT/LCT 20%). Applying the Box–Behnken design and response surface methodology allowed for the optimization of the formulation in terms of LE% and particle size (MDD). The optimal honokiol-loaded nanoemulsion was characterized by physicochemical parameters sufficient for parenteral administration with a satisfactory effect on glioblastoma cell lines. The long-term and short-term stress studies showed the physicochemical stability of the formulation if the appropriate storage conditions are used. The honokiol-loaded nanoemulsion is a promising pharmaceutical formulation for further development in the adjuvant therapy of glioblastoma.

## Figures and Tables

**Figure 1 pharmaceutics-15-00448-f001:**
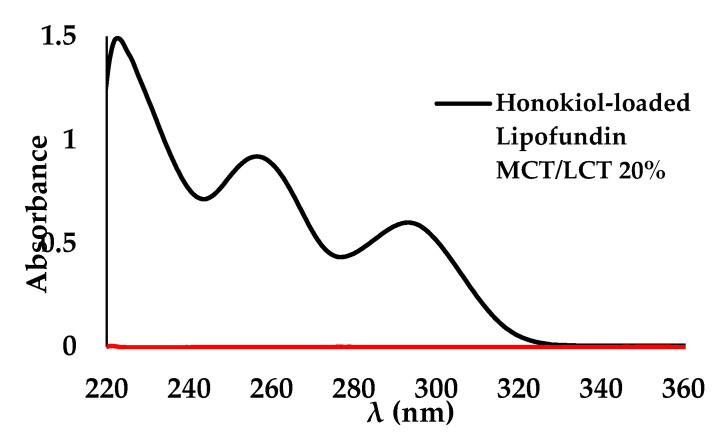
UV-VIS spectra of honokiol-loaded Lipofundin MCT/LCT 20% and Lipofundin MCT/LCT 20%.

**Figure 2 pharmaceutics-15-00448-f002:**
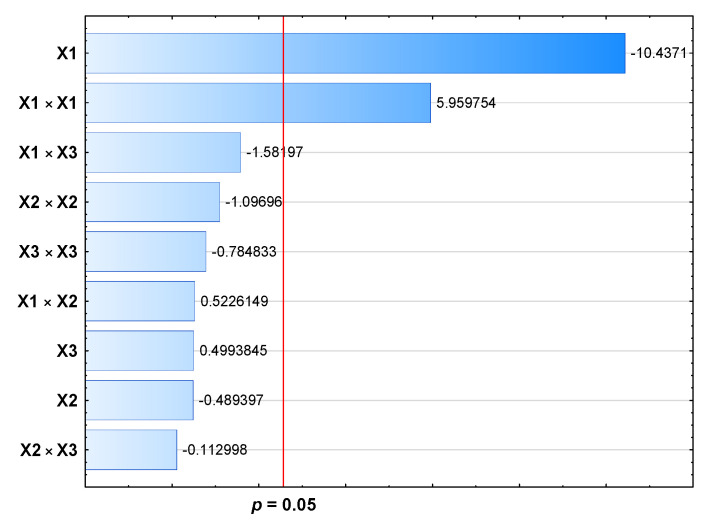
Pareto chart for the BBD for particle size.

**Figure 3 pharmaceutics-15-00448-f003:**
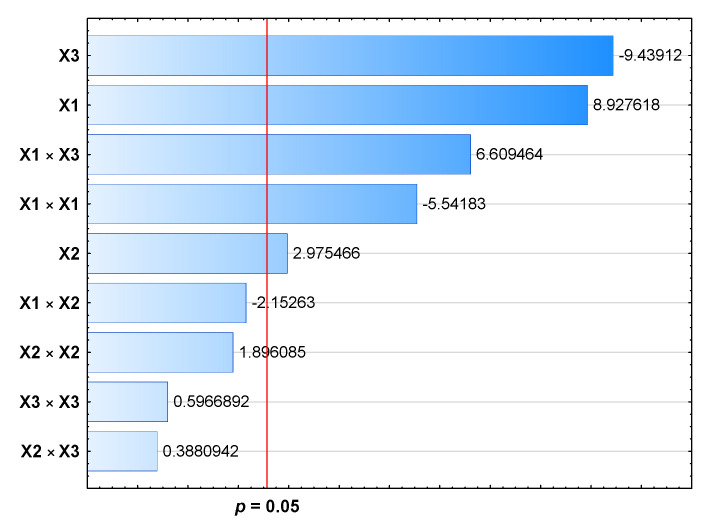
Pareto chart for the BBD for loading efficiency.

**Figure 4 pharmaceutics-15-00448-f004:**
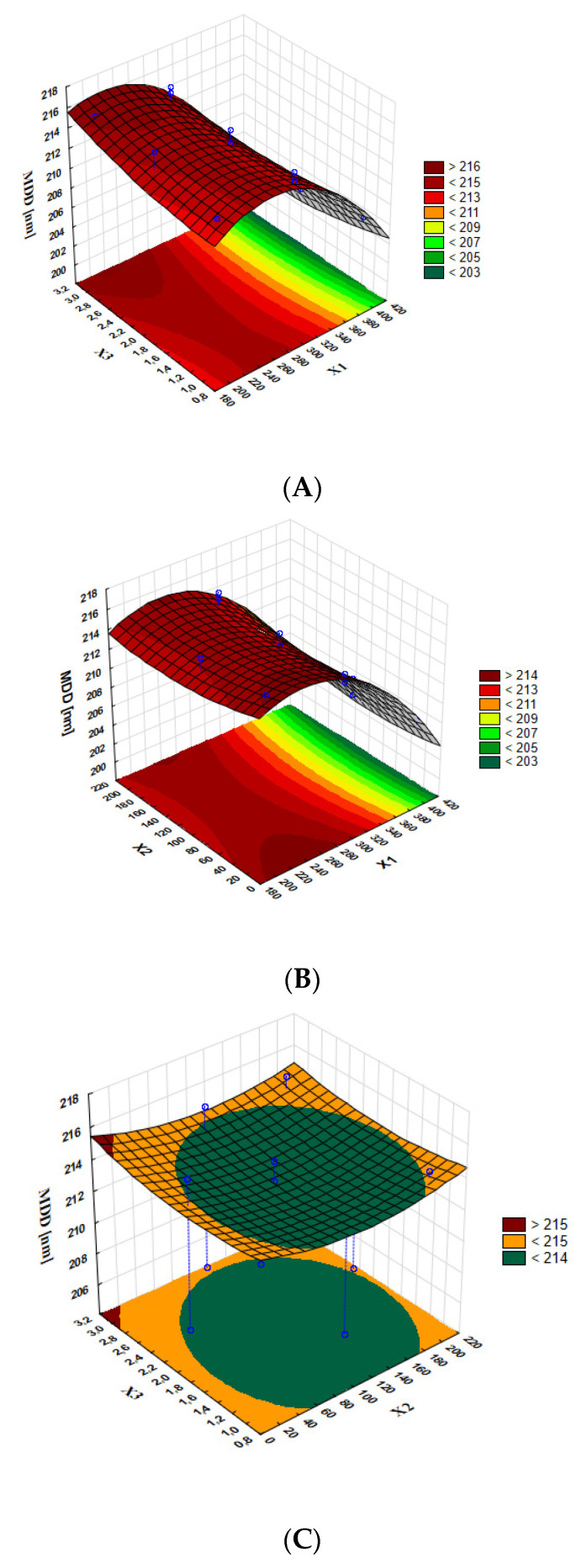
Response surface plots present the interaction effect of (**A**) concentration of honokiol (X3) and shaking speed (X1); (**B**) time of shaking (X2) and shaking speed (X1); and (**C**) concentration of honokiol (X3) and time of shaking (X2) on particle size.

**Figure 5 pharmaceutics-15-00448-f005:**
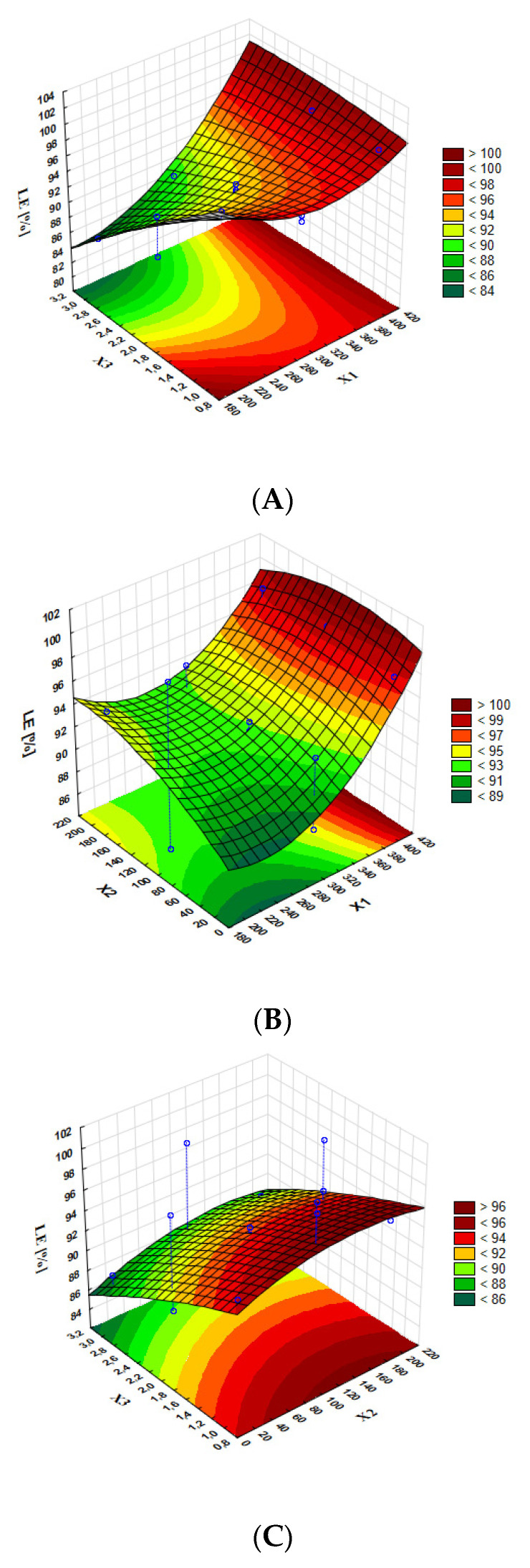
Response surface plots present the interaction effect of (**A**) concentration of honokiol (X3) and shaking speed (X1); (**B**) time of shaking (X2) and shaking speed (X1); and (**C**) concentration of honokiol (X3) and time of shaking (X2) on loading efficiency.

**Figure 6 pharmaceutics-15-00448-f006:**
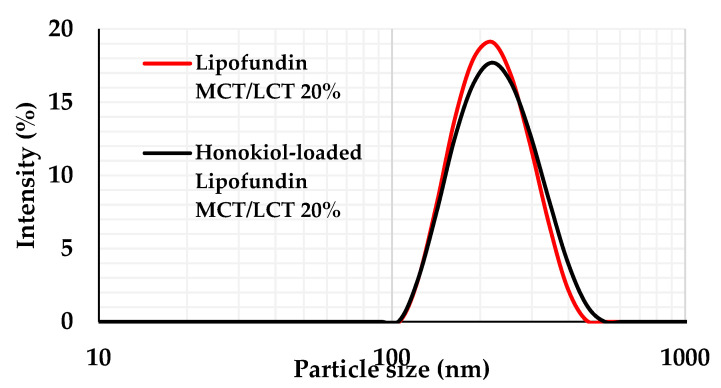
Particle size distribution of Lipofundin MCT/LCT 20% and optimal formulation of honokiol-loaded Lipofundin MCT/LCT 20%.

**Figure 7 pharmaceutics-15-00448-f007:**
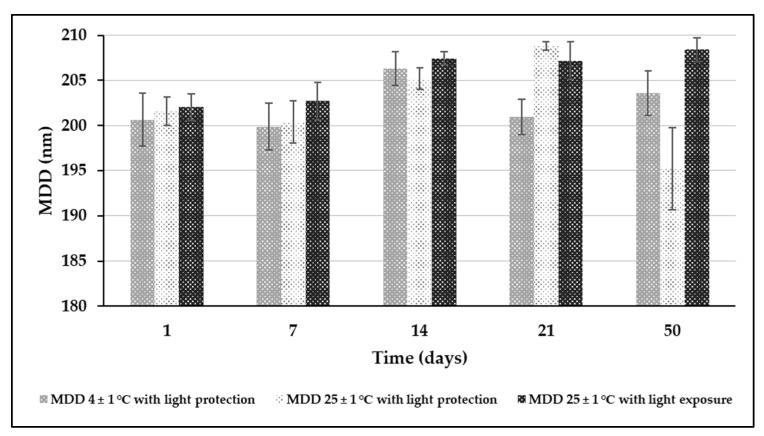
Results of particle size determination in long-term stability test of honokiol-loaded nanoemulsion.

**Figure 8 pharmaceutics-15-00448-f008:**
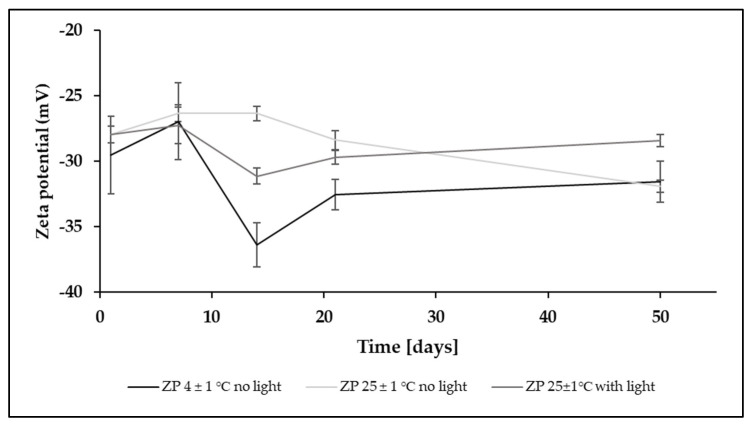
Results of zeta potential determination in long-term stability test of honokiol-loaded nanoemulsion.

**Figure 9 pharmaceutics-15-00448-f009:**
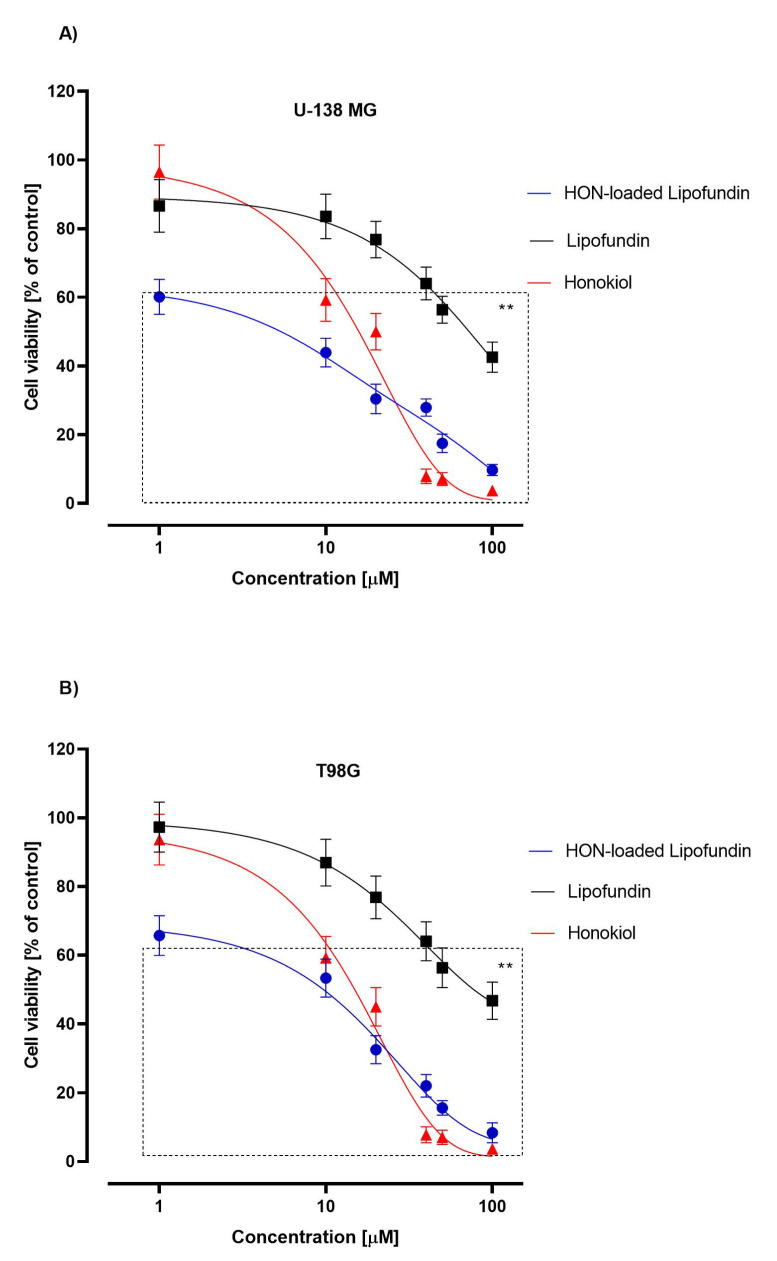
The cytotoxicity evaluation of honokiol-loaded Lipofundin MCT/LCT 20%, Lipofundin MCT/LCT 20%, and honokiol in U-138 MG (**A**) and T98G (**B**) cell lines, based on 24 h MTT test. DMSO-treated cells (control) were assigned as 100% cell viability. The mean values SEM from three independent experiments with four measurements per assay are presented. **—*p* < 0.05.

**Table 1 pharmaceutics-15-00448-t001:** Experimental matrix of randomized runs in Box–Behnken design.

Formulation Code	X1 Shaking Speed (rpm)	X2 Shaking Time (min)	X3 HON Concentration (mg/mL)
F1	200	15	2
F2	400	15	2
F3	200	195	2
F4	400	195	2
F5	200	105	1
F6	400	105	1
F7	200	105	3
F8	400	105	3
F9	300	15	1
F10	300	195	1
F11	300	15	3
F12	300	195	3
F13	300	105	2
F14	300	105	2
F15	300	105	2

**Table 2 pharmaceutics-15-00448-t002:** Characteristics of studied honokiol-loaded formulations.

Formulation Code	LE% (SD) [%]	MDD (SD) [nm]	PDI (SD)	ZP (SD) [mV]
F1	88.52 (0.05)	215.8 (1.5)	0.07 ( 0.04)	−31.2 (0.6)
F2	97.85 (2.40)	206.3 (1.5)	0.07 (0.02)	−28.0 (0.2)
F3	93.78 (5.33)	213.6 (0.8)	0.07 (0.02)	−27.7 (0.8)
F4	98.75 (0.52)	205.4 (2.0)	0.09 (0.02)	−27.1 (0.3)
F5	99.77 (3.90)	213.4 (2.1)	0.09 (0.02)	−27.1 (0.1)
F6	98.71 (0.88)	206.7 (1.1)	0.08 (0.00)	−30.8 (0.5)
F7	85.41 (1.09)	215.3 (1.0)	0.08 (0.01)	−27.9 (0.3)
F8	97.74 (2.91)	204.9 (1.1)	0.10 (0.02)	−28.1 (0.3)
F9	94.02 (8.05)	213.6 (2.1)	0.04 (0.04)	−26.7 (0.3)
F10	94.81 (7.17)	214.5 (0.8)	0.08 (0.02)	−26.9 (0.1)
F11	87.78 (2.23)	214.5 (1.6)	0.09 (0.03)	−27.8 (0.2)
F12	89.35 (2.58)	215.1 (1.9)	0.07 (0.02)	−27.4 (0.9)
F13	92.62 (3.55)	214.6 (0.5)	0.09 (0.01)	−30.1 (0.5)
F14	92.40 (4.74)	211.8 (2.2)	0.07 (0.02)	−25.7 (0.2)
F15	93.38 (1.42)	213.3 (0.8)	0.09 (0.02)	−25.3 (0.4)

SD—standard deviation.

**Table 3 pharmaceutics-15-00448-t003:** Results of ANOVA test for particle size.

Source	dF	SS	MS	F-Values	*p*-Values	*p*-Values
**Model**	9	210.879	23.431	16.83	0.003	< 0.05
**Linear model**	3	152.351	50.784	36.47	0.001	< 0.05
**X1**	1	151.670	151.670	108.93	0.000	< 0.05
**X2**	1	0.333	0.333	0.24	0.645	> 0.05
**X3**	1	0.347	0.347	0.25	0.639	> 0.05
**Quadratic model**	3	54.646	18.215	13.08	0.008	< 0.05
**X1 × X1**	1	49.453	49.453	35.52	0.002	< 0.05
**X2 × X2**	1	1.675	1.675	1.20	0.323	> 0.05
**X3 × X3**	1	0.858	0.858	0.62	0.468	> 0.05
**Interactions**	3	3.882	1.294	0.93	0.491	> 0.05
**X1 × X2**	1	0.380	0.380	0.27	0.624	> 0.05
**X1 × X3**	1	3.484	3.484	2.50	0.175	> 0.05
**X2 × X3**	1	0.018	0.018	0.01	0.914	> 0.05
**Error**	5	6.962	1.392			
**Lack of fit**	3	3.030	1.010	0.51	0.713	> 0.05
**Pure error**	2	3.932	1.966			
**Total**	14	217.841				
**Regression equation**	MDD [nm] = 191.36 + 0.1911 X1 − 0.0285 X2 + 0.116 X3 − 0.000366 X1 × X1 + 0.000083 X2 × X2 + 0.00482 X3 × X3 + 0.000034 X1×X2 − 0.000933 X1 × X3 − 0.000074 X2 × X3					

**Table 4 pharmaceutics-15-00448-t004:** Results of ANOVA test for loading efficiency.

Source	dF	SS	MS	F-values	*p*-Values	*p*-Values
**Model**	9	269.446	29.9385	29.22	0.001	<0.05
**Linear model**	3	182.006	60.6685	59.22	0.000	<0.05
**X1**	1	81.655	81.6552	79.70	0.000	<0.05
**X2**	1	9.070	9.0703	8.85	0.031	<0.05
**X3**	1	91.280	91.2800	89.10	0.000	<0.05
**Quadratic model**	3	37.784	12.5946	12.29	0.010	<0.05
**X1 × X1**	1	31.464	31.4644	30.71	0.003	<0.05
**X2 × X2**	1	3.683	3.6832	3.60	0.116	>0.05
**X3 × X3**	1	0.365	0.3648	0.36	0.577	>0.05
**Interactions**	3	49.657	16.5523	16.16	0.005	<0.05
**X1 × X2**	1	4.747	4.7473	4.63	0.084	>0.05
**X1 × X3**	1	44.755	44.7554	43.69	0.001	<0.05
**X2 × X3**	1	0.154	0.1543	0.15	0.714	>0.05
**Error**	5	5.123	1.0245			
**Lack of fit**	3	4.597	1.5322	5.83	0.150	>0.05
**Pure error**	2	0.526	0.2629			
**Total**	14	274.569				
**Regression equation**	LE [%] = 129.10 − 0.1974 X1 + 0.0697 X2 − 1.238 X3 + 0.000292 X1 × X1 − 0.000123 X2 × X2 − 0.00314 X3 × X3 − 0.000121 X1 × X2 + 0.003345 X1 × X3 + 0.000218 X2 × X3					

**Table 5 pharmaceutics-15-00448-t005:** Short-term stress stability studies of honokiol in honokiol-loaded nanoemulsion.

Time	No Stress Condition (4 ± 2 °C, No Light Exposure)	Oxidative Stress Condition (25 ± 2 °C, No Light Exposure)	High-Temperature Condition (60 ± 1 °C, No Light Exposure)	Accelerated-Light Condition (35 ± 2 °C)
**t = 0 h**	100.00	100.00	100.00	100.00
**t = 24 h**	100.00 (0.47)	99.92 (0.96)	96.37 (3.62)	91.72 (1.86) ^a^
**t = 48 h**	100.12 (3.05)	99.76 (1.84)	97.25 (2.08)	99.05 (1.21) ^b^
**t = 72 h**	99.33 (1.29)	99.15 (2.21)	97.70 (1.34)	-

^a^—result obtained after 22 h for studied sample. ^b^—result obtained after 22 h for reference sample.

## Data Availability

The data is contained in the article.

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
