# Peer review of "Honokiol-Loaded Nanoemulsion for Glioblastoma Treatment: Statistical Optimization, Physicochemical Characterization, and an In Vitro Toxicity Assay"

_pharmaceutics, 2023, doi:10.3390/pharmaceutics15020448_

Round 1
Reviewer 1 Report
Dear Authors,
I found this manuscript interesting, but I have several comments:
- According to what criteria were the limits of shaking speed, duration of shaking, and concentration of honokiol selected? A justification would be required. Lines 125–126
- I suggest choosing the same measurement units (nm or µm). Line 145
- According to the European Pharmacopoeia, it is purified water, not distilled water. Line 168
- Did the nanoemulsion without honokiol not show absorbance at 356 nm wavelength? It would be good if you put the spectra in the method: nanoemulsions without honokiol and honokiol-loaded nanoemulsions. Lines 170–183
- Was the wavelength 356 nm or 256 nm? Lines 178, 209
- I suggest using mean (standard deviation) instead of mean ± standard deviation since ± is used only with standard error. Lines 244–245
- Were statistical analysis methods used to compare the results?
- You write "However, the polydispersity index results show that all prepared honokiol-loaded nanoemulsions were homogeneous, as the PDI was below 0.7 in each case". Why is the PDI limit 0.7? For example, PDI should not exceed 0.3 for microemulsions. Lines 284–286
- Why are insignificant variables included in regression equations (Tables 3 and 4)?
- You write "The optimal and, at the same time, the lowest MDD was obtained when the shaking speed was about 200 rpm". However, according to the three-dimensional graphs of the variables and Table 2, the lowest MDD was when the shaking speed was about 400 rpm. Please comment. Lines 333–334
- To optimize the nanoemulsion production process, how were the variables (MDD, LE%) characterized (minimum, maximum, etc.) and ranked according to the importance to obtain the optimal parameters? What was the desirability value of the obtained optimal parameters? Line 349
- Was the selected Honokiol-loaded nanoemulsion produced according to optimal parameters? Lines 352
- Comment, why Lipofundin (nanoemulsion without honokiol) decreased cell viability at higher concentrations (Figure 7)?
- Are in vivo studies planned?
- You write "…the oil-in-water system itself make the developed honokiol-loaded nanoemulsion a promising drug delivery system for glioblastoma tumors". I think it is too bold to say that until in vivo studies are done. Lines 481–483
- Why were pH and OSM 15 experimental design formulations not evaluated and included in the search for optimal nanoemulsion production parameters?
- You write "The Lipofundin MCT/LCT 20% produced only a slightly toxic effect towards both cell lines (Fig. 7)". However, cell viability was only 50% at the highest concentration. I don't think this is a slightly toxic effect. Line 564
Author Response
Please find our response in the attached file.

Reviewer 2 Report
The comments for research work on" Honokiol-loaded nanoemulsion for glioblastoma treatment: statistical optimization, physicochemical characterization, and an in vitro toxicity assay".
1. As mentioned, Lipofundin MCT/LCT 20% is used as a nanoemulsion, which is readily available. Why used ready-to-use nanoemulsion?
2. Is the drug compatible with Lipofundin MCT/LCT 20%? If any preformulation study is done please share.
3. The optimization is done for the loading of Honokiol, by varying the shaking speed and time. The asking time variation taken is very wide from 115, 105 to 195 mins. Why? What kind of shaker used.
4. Why high speed homogenizer not used as it can give high dispersion in minimum time?
5. Cite the method used for manufacturing as well as characterize if they are not novel.
6. Honokiol, can the author briefly describe its property for better understanding?
7. How author so sure that all Honokiol, is loaded or not, any test for free Honokiol carried out?
8. Please share the imaging technology photo graph for shape and size explanation. (TEM would be great)
Author Response
Please find enclosed our response in the attached file.

Round 2
Reviewer 1 Report
Dear Authors,
Thank you for your answers to my questions and observations. However, I disagree with some of your answers:
- Did the nanoemulsion without honokiol not show absorbance at 356 nm wavelength? It would be good if you put the spectra in the method: nanoemulsions without honokiol and honokiol-loaded nanoemulsions. Lines 170–183
Before collecting the UV-Vis spectra of samples containing honokiol, the blank samples (without the addition of honokiol) were prepared according to the same method. All UV-Vis measurements were conducted with respect to the intensity of light passing through a blank sample. Such an approach assures that the background of the solvent and solutes that are not adding their spectra to the sample spectrum.
We add an appropriate explanation in the Materials and Methods section. Lines 187-190
Comment: I still suggest presenting the UV spectra of both the blank samples (without the addition of honokiol) and the honokiol-loaded nanoemulsion.
2. I suggest using mean (standard deviation) instead of mean ± standard deviation since ± is used only with standard error. Lines 244–245
Corrected
Comment: In both the text and tables, the standard deviation is not shown in parentheses.
3. What statistical analysis did you use to evaluate the results: a) Short-term and long-term stability studies; b) In Vitro Cytotoxicity Studies?
4. Why are insignificant variables included in regression equations (Tables 3 and 4)?
The regression equation was prepared using the Statistica 10, StatSoft package based on the obtained results. The statistical method (Box Behnken design) used assumes the presentation of the regression equation on the basis of all variables, including those that are statistically insignificant.
Comment: However, the presence of insignificant variables in regression equations reduces the predictive accuracy of such equations.
5. You write "The optimal and, at the same time, the lowest MDD was obtained when the shaking speed was about 200 rpm". However, according to the three-dimensional graphs of the variables and Table 2, the lowest MDD was when the shaking speed was about 400 rpm. Please comment. Lines 333–334
Thank you very much for this comment and attention. Of course, the Reviewer is right. There was a logical error here. As the indicated MDD value is satisfactorily low (below 220 nm) for a shaking value of 200 rpm, there was an error in the description of the results.
This sentence has been corrected as follows (lines 346-347):
The optimal and, at the same time, the lowest MDD was obtained when the shaking speed was about 200 rpm.
Comment: I do not agree with your corrections and I suggest you take a good look at Table 2 (the lowest MDD was obtained when the shaking speed was about 400 rpm – F2, F4, F6, F8).
6. You did not answer this question: What was the desirability value (optimization gives it between 0 and 1) of the obtained optimal parameters?
7. If Lipofundin (nanoemulsion without honokiol) reduces the viability of tumor cells, why should honokiol be added? In this case, do you think Lipofundin (nanoemulsion without honokiol) would not affect healthy cells?
8. Your answer: The low toxicity of Lipofundin MCT/LCT 20% was observed in low concentrations up to 10 uM. Therefore, we rewrite this sentence as follows (lines 602-611): The Lipofundin MCT/LCT 20% produced only a slightly toxic effect in low concentrations up to 10 uM towards both cell lines, with the viability over 80% (Fig. 7). This might be attributed to the presence of biocompatible ingredients in this nanoemulsion i.e. refined soybean oil, medium-chained triglycerides, purified egg phospholipids, glycerol, sodium oleate, and α-tocopherol, that in low concentration are not toxic for cells.
Comment: The first and last sentences contradict each other. I think a 20% decrease in cell viability means there is toxicity.
